# New Advances in Biomedical Application of Polymeric Micelles

**DOI:** 10.3390/pharmaceutics14081700

**Published:** 2022-08-15

**Authors:** Ana Figueiras, Cátia Domingues, Ivana Jarak, Ana Isabel Santos, Ana Parra, Alberto Pais, Carmen Alvarez-Lorenzo, Angel Concheiro, Alexander Kabanov, Horacio Cabral, Francisco Veiga

**Affiliations:** 1Univ Coimbra, Laboratory of Drug Development and Technologies, Faculty of Pharmacy, 3000-548 Coimbra, Portugal; 2Univ Coimbra, REQUIMTE/LAQV, Group of Pharmaceutical Technology, 3000-548 Coimbra, Portugal; 3Univ Coimbra, Faculty of Medicine, Institute for Clinical and Biomedical Research (iCBR), Area of Environment Genetics and Oncobiology (CIMAGO), 3000-548 Coimbra, Portugal; 4Coimbra Chemistry Center (CQC), Faculty of Science and Technology, University of Coimbra, 3004-531 Coimbra, Portugal; 5Departamento de Farmacología, Farmacia y Tecnología Farmacéutica, I+D Farma (GI-1645), Facultad de Farmacia, iMATUS, and Health Research, Institute of Santiago de Compostela (IDIS), Universidade de Santiago de Compostela, 15782 Santiago de Compostela, Spain; 6Eshelman School of Pharmacy, UNC Chapel Hill, Chapel Hill, NC 27599, USA; 7Department of Bioengineering, Graduate School of Engineering, The University of Tokyo, 7-3-1 Hongo, Bunkyo-ku, Tokyo 113-8656, Japan; 8Innovation Center of NanoMedicine (iCONM), Kawasaki Institute of Industrial Promotion, 3-25-14 Tonomachi, Kawasaki-ku, Kawasaki 210-0821, Japan

**Keywords:** polymeric micelles, copolymers, cancer-target delivery, nanocarrier, biomedical applications, theranostic

## Abstract

In the last decade, nanomedicine has arisen as an emergent area of medicine, which studies nanometric systems, namely polymeric micelles (PMs), that increase the solubility and the stability of the encapsulated drugs. Furthermore, their application in dermal drug delivery is also relevant. PMs present unique characteristics because of their unique core-shell architecture. They are colloidal dispersions of amphiphilic compounds, which self-assemble in an aqueous medium, giving a structure-type core-shell, with a hydrophobic core (that can encapsulate hydrophobic drugs), and a hydrophilic shell, which works as a stabilizing agent. These features offer PMs adequate steric protection and determine their hydrophilicity, charge, length, and surface density properties. Furthermore, due to their small size, PMs can be absorbed by the intestinal mucosa with the drug, and they transport the drug in the bloodstream until the therapeutic target. Moreover, PMs improve the pharmacokinetic profile of the encapsulated drug, present high load capacity, and are synthesized by a reproducible, easy, and low-cost method. In silico approaches have been explored to improve the physicochemical properties of PMs. Based on this, a computer-aided strategy was developed and validated to enable the delivery of poorly soluble drugs and established critical physicochemical parameters to maximize drug loading, formulation stability, and tumor exposure. Poly(2-oxazoline) (POx)-based PMs display unprecedented high loading concerning water-insoluble drugs and over 60 drugs have been incorporated in POx PMs. Among various stimuli, pH and temperature are the most widely studied for enhanced drug release at the site of action. Researchers are focusing on dual (pH and temperature) responsive PMs for controlled and improved drug release at the site of action. These dual responsive systems are mainly evaluated for cancer therapy as certain malignancies can cause a slight increase in temperature and a decrease in the extracellular pH around the tumor site. This review is a compilation of updated therapeutic applications of PMs, such as PMs that are based on Pluronics^®^, micelleplexes and Pox-based PMs in several biomedical applications.

## 1. Introduction

Polymer-based nanosystems are engaging platforms in different targeted therapies. The increased interest in integrating nanotechnology with cancer diagnosis and treatment has led to the development of various tumor-targeting nanoparticles for cancer applications [1,2,3,4,5]. Currently, the use of nanomedicines to entrap conventional therapies and diagnostic tools has been implemented to address a synergistic strategy to overcome metastatic cancers [6,7,8]. In this sense, polymeric micelles (PMs) have shown great potential in the delivery of lipophilic drugs, small molecules, and proteins for cancer treatment, and potential benefits to carrier imaging moieties aiming cancer theranostics [9]. Furthermore, their application in dermal drug delivery is also relevant and will be described in more detail in this article.

PMs are drug delivery nanosystems that are characterized by a core-shell structure, that is developed through the self-assembly of amphiphilic block copolymers in an aqueous solution. In this way, it is possible for researchers to explore various polymeric combinations, differing in functionalization and flexible chemistry [10]. Some of the widely accepted hydrophobic copolymers that are used in PMs are: (i) poly(caprolactone) (PCL), (ii) poly(lactide) (PLA), (iii) polyesters, poly(lactide-co-glycolide) (PLGA), (iv) lipids, and (v) poly(amino acids) [11]. The hydrophobic part of block copolymers is intended to solubilize poorly soluble drugs in the core as well as release the drug from the PMs. Furthermore, hydrophobic interactions between the drug and hydrophobic unit of copolymers are very well recognized as one important factor in solubilizing the drugs in PMs as well as retarding the release rate of the drug to the external solution [12,13,14]. Several hydrophobic copolymers have been developed and evaluated as core-forming blocks in PMs and show the capacity to solubilize poorly soluble drugs, as demonstrated in Table 1 [15]. Hydrophilic copolymers are commonly used to wrap the hydrophobic core in PMs, such as: (i) poly(oxazolines), (ii) poly(ethylene glycol), (iii) chitosan, (iv) hyaluronic acid, as well as (v) dextran [11,16]. The functionalities of the hydrophilic shells were widely studied and according to those studies (Table 1), the physicochemical properties of hydrophilic polymers, such as surface density and molecular weight, were closely related to the systemic circulation time, biodistribution, and stability of PMs in vivo [17,18].

Regarding the synthesis of PMs, in a diluted aqueous solution, polymers (amphiphilic units) will be presented as dispersed polymer units in a medium and amphiphiles work as surfactants reducing the surface tension at the air-water interface. More units are added to the system until the aggregation of polymers occurs in an ordered structure, due to the saturation of the bulk point. This concentration is known as the critical micelle concentration (CMC) [16]. Thus, CMC is defined as the minimum concentration of polymers in solution leading to micelle formation. Self-assembly of amphiphilic polymers (i.e., hydrophobic and hydrophilic units) results in micelles (polymeric nanoparticles), where copolymer concentrations are above to CMCs [19]. Consequently, the CMC value is one of the most important parameters to define micelles’ thermodynamic stability [16]. The CMC of polymers can be analyzed by surface-tension measurements as well as conductivity [20] and light-scattering methods [21,22]. Additionally, the Ostwald ripening phenomena is essential in forming stable PMs/micelles. Ostwald ripening defines the process in which larger particles grow by “devouring” smaller particles, which may contribute to the instability of micelles. The driving force for Ostwald ripening arises because the concentration of a drug in the vicinity of small particles is greater, and that of large particles is less than the average supersaturation [23]. Depending on the hydrophilic and hydrophobic units as well as solvent conditions, the morphologies of micelles take different shapes: (i) spheres, (ii) tubules, (iii) inverse micelles, (iv) bottle-brush shapes, and so on [24]. 

The development and formulation of PMs can be carried out by different methods, such as solvent evaporation [25], dilution [26], lyophilization [27], dialysis [28], and oil-in-water emulsion.

The main advantage of PMs compared to other low molecular weight (MW) surfactant micelles is that the first has a lower CMC index and, therefore, higher kinetic stability. In fact, for low MW surfactant micelles, disassembly occurs in the range of microseconds, whereas in PMs the drug can also be covalently bound to the polymer (polymer-drug conjugate), making it a double weapon in therapy [16].

To date, there are three generations of PMs delivery systems for drug targeting: (i) passive PMs, (ii) active PMs, and (iii) multifunctional PMs, where each subsequent generation exhibits greater efficiency and better tissue specificity [29]. A new generation of more effective PMs evolved due to the emergence of stimuli-responsive polymers. Many advantages of PMs have been proven with their unique core-shell architecture. The most significant advantage of PMs compared with other nanosystems, including lipid-based nanoparticles, is their small size, between 10 and 100 nanometers, and the presence of a micelle corona (as described in more detail in Table 2). These features provide PMs an effective steric protection for the micelles and determine the micelle hydrophilicity, charge, length, and surface density of hydrophilic blocks [30]. Furthermore, due to their small size, PMs [31] can be absorbed by the intestinal mucosa with the drug, and they transport the drug in the bloodstream until the therapeutic target. Moreover, PMs are difficult to recognize by the macrophage phagocytic system, increasing their circulation half-time [32,33]. In general, PMs can improve the pharmacokinetic profile of the encapsulated drug, present high load capacity [34], and are synthesized by a reproducible, easy, and low-cost method [35].

The formulation of PMs should be performed in a fit-by-design perspective encompassing specific regulatory requirements, minimizing risks, and promoting safety and efficacy [40]. Based on this, a Reflection Paper was elaborated by the European Medicines Agency (EMA) with the main features for the development of block-copolymer micelles intended to be used for medicinal purposes(Figure 1) [16].

The main goal of this review is to compile several advanced applications of PMs presented by renowned international researchers under the scope of the conference “1st Advances in Polymeric Micelles for therapeutic applications”, which elapsed at the Faculty of Pharmacy of the University of Coimbra, Portugal. 

## 2. Polymeric Micelles as Versatile Drug Delivery Carriers

Amphiphilic copolymers are highly versatile components of drug delivery systems [41,42,43,44,45,46]. Tuning the hydrophilic and the hydrophobic blocks allow obtaining a variety of self-assembled structures with the different capabilities of hosting drugs, enhancing their apparent solubility and stability, crossing biological barriers, and delivering the drugs to the right site [47,48,49,50]. Also, some block copolymers have been shown to act as active molecules that are able to enhance the therapeutic effects of the drugs (Figure 2). The combination of block copolymers with cyclodextrins expands, even more, the spectrum of applications in the drug delivery field. Poly(pseudo)rotaxane formation enables fine-tuning of the rheological properties of the formulations and the drug release kinetics [51,52,53].

## 3. Morphology, Partitioning, and Pharmacological Performance in Polymeric Micelles

Poly(2-oxazoline) (POx)-based PMs display unprecedented high loading concerning water-insoluble drugs. Such PMs greatly enhance the solubility and stability of drugs and improve their efficacy and safety in a transformative way. Over 60 drugs and drug combinations have been incorporated in POx PMs. We developed and validated a computer-aided strategy to enable the delivery of poorly soluble drugs and established critical physicochemical parameters to maximize drug loading, formulation stability, and tumor exposure [54]. The models predicting drug loading efficiency (LE) and loading capacity (LC) using novel descriptors of drug-polymer complexes were employed for the virtual screening of drug libraries. The micelle morphology impacts on the drug pharmacological performance in spherical and worm-like particles [16]. PMs can elongate over time from spherical to worm-like structures, depending on the amount and type of medication. Tiny spherical micelles rapidly accumulate in tumors while carrying more drugs than the worm-like micelles [55]. As a result, greater antitumor effects are seen with spherical micelles. The dynamic character of drug–micelle species and the control of micelle morphology play a critical role in the drug delivery in tumors [56,57]. Using a rational computer-aided approach, POx micelles that were loaded with chemosensitizers, anticancer agents, and immunomodulating small molecules were designed to improve the treatment of non-small cell lung cancer (NSCLC) [58] (Figure 3), triple-negative, and other condition cancers. Superior antitumor activity of the co-loaded two-drug micelles compared to single drug micelles or their combination as well as free drug combination was demonstrated using several animal models (Figure 3B,C) [59]. Hence, a powerful possibility of “drug design by co-formulation” as a therapeutic platform approach has been outlined [60]. The POx PMs approach has shown significant promise for the delivery and co-delivery of agents that target the tumor microenvironment (TME) as a strategy for cancer immunotherapy. Using this approach, small molecules target tumor-associated macrophages (TAMs), tumor-associated fibroblasts (TAFs), myeloid-derived-suppressor cells (MDSCs), and regulatory T-cells (Tregs) have been delivered to enhance the adaptive T-cell immunity and improve the outcomes of immunologically cold cancers [61,62,63,64]. These innovations address the need for accelerating the translation of novel PMs for therapeutic applications and drug products for cancer and other diseases [54,59,65,66,67].

## 4. pH-Sensitive Polymeric Micelles for Tumor-Targeted Delivery of Proteins

Protein drugs, including cytokines, enzymes, and antibodies, have demonstrated great potential as therapeutics [68,69]. However, they usually present several issues, such as poor pharmacokinetics, metabolic stability, and adverse immune responses [70,71,72]. Nanomedicines have a high potential for developing innovative delivery approaches for overcoming the limitations of these agents, as well as providing enhanced targeting capabilities [73,74].

The current therapeutic proteins that are used in the clinic are aimed at targets thata re expressed on the cellular membrane or outside of the cells. This is because proteins are impermeable to the cell membrane, which limit their application for reaching intracellular targets. For improving the stability and reducing the immunogenicity of such types of therapeutic proteins, the most common approach is the modification with PEG chains. PEGylated proteins can also alter the pharmacokinetics, providing more favorable clearance profiles and tissue distributions [75]. We have recently applied a PEGylation approach for modifying antibodies that were directed to programmed death ligand 1 (PD-L1) and promoting their accumulation in brain tumors (Figure 4) [76]. In fact, brain tumors are highly impermeable to drugs, proteins, and nanomedicines due to the presence of the tight blood-brain tumor barrier (BBTB) [77,78,79,80]. In our study, the modification of the anti-PD-L1 antibodies with multiple glycosylated PEG chains *via* reducible bonds allowed the engagement with the glucose transporter 1 (GLUT-1) that was expressed on the endothelial cells forming the BBTB. Then, the anti-PD-L1 antibodies were actively transported into brain tumors, probably through a GLUT-1 recycling mechanism, as observed for other glycosylated nanomedicines [81,82]. Thus, the accumulation in the brain tumor of was 10-fold higher for the glycosylated-PEG-conjugated antibodies that for the native antibodies [76]. Moreover, the glycosylated-PEG-conjugated antibodies selectively accumulated in the tumors, but not in healthy brain. Such high and selective accumulation in the brain tumors allowed the eradication of glioblastoma models that are resistant to anti-PD-L1 therapy without inducing immune-related adverse effects [83,84].

When the therapeutic protein is directed to intracellular targets, PEGylation is not sufficient for surmounting the cellular membrane, the endosomal membranes and, in some cases, the nuclear membrane, which are formidable barriers for delivery of drugs [85,86]. Thus, for overcoming these barriers, proteins could be modified with polymers capable of promoting the access through these membranes [87,88]. A common approach is to use polymers with membrane disrupting capability, such as systems having hydrophobic moieties and cationic groups [89]. Moreover, given the lower pH of endosomes (pH 6.5–4.5) compared to the extracellular physiological pH (pH 7.4), it is possible to design carriers selectively disrupting endosomal membranes. For example, we have used polymer micelles that were prepared from PEG-poly(aspartate-diethylenetriamine) (PEG-poly(Asp(DET))) block copolymers, which can break endosomes due to their buffering capacity at endosomal pH [90], for delivering charge-converted antibodies inside the cells [91]. In this case, the antibodies were directed to target the nuclear pore complex after recovering their charge in the endosome and escaping into the cytosol. It is worth noting that the current paradigm of nanocarriers that are aimed at intracellular macromolecule delivery is to maximize their ability to disrupt endosomal membranes for effectively escaping from endosomes [92,93,94]. However, such endosomal escape strategies lack cellular specificity. It may be possible that nanocarriers designed to discriminate the endosomes of target cells could provide specific transport of their payloads into the cytosol of these particular cells, which could result in highly targeted intracellular therapies [95]. Research in this direction is ongoing in our group [96].

## 5. Pluronic^®^-Based Polymeric Micelles

Amphiphilic block copolymers have the spontaneous ability to form a micellar structure in an aqueous medium, presenting a hydrophilic crown and a hydrophobic core [97,98,99]. In this way, through hydrophobic interactions, the encapsulation of poorly water-soluble drugs is possible. Among the block copolymers, the ones that have raised the most scientific interest in the elaboration of PMs are poloxamers and poloxamines [73,100,101,102,103]. Poloxamers or triblock copolymers, also known as Pluronics^®^ (Figure 5), are the most used and studied polymers to develop PMs. Pluronics^®^ are U.S. Food and Drug Administration-approved triblock copolymers, which are widely used for solubilization and delivery of drugs, gene therapy, diagnostics, and tissue engineering applications [104,105,106]. Their unique physicochemical and biological properties depend on their MW and/or on the size of the hydrophobic, poly(propylene oxide), core and the hydrophilic, poly(ethylene oxide), side chains (PEO_n_-PPO_m_-PEO_n_). Pluronics^®^ can self-assemble into PMs in an aqueous medium above their CMC/CMT [107].

Poloxamines, also known as Tetronics^®^, present an X-shape (Figure 5), with a central ethylenediamine unit, four arms of PPO, and PEO blocks. Compared to Pluronics^®^, these polymers have a more versatile and unique structure that gives them the capacity to respond to multiple stimuli.

### 5.1. Pluronics^®^ for Cancer Treatment

The capacity to self-aggregate into a variety of thermodynamically-stable nanosized supramolecular structures has been exploited to overcome the problem of low drug water solubility and has been widely used to deliver a variety of hydrophobic antitumor chemotherapeutics [108]. While the augmented hydrophobic surface of polymers is crucial for the successful solubilization of hydrophobic drugs, the outer hydrophilic shell that is composed of highly polar and flexible PEO chains provides stability and stealth properties to Pluronic micelles by preventing aggregation and interactions with mononuclear phagocytic systems and other biological compartments that lead to the removal of micelles from systemic circulation [109]. Binary micelles that are composed of Pluronics^®^ with complementary polarities emerged as improved formulations with superior thermodynamic and kinetic properties with higher loading capacity and stability [110]. L61/F127 mixed micelles that were loaded with doxorubicin presented the first Pluronic^®^-based micellar formulation to enter clinical trials (SP1049C, Supratek Pharma Inc., Dorval, PQ, Canada) [111]. On the other hand, the release of hydrophobic drugs that were localized within the hydrophobic core of undegradable Pluronic^®^ micelles depends on the diffusion rate that is dictated by the strength of the interactions with the core as well as micellar stability. Due to the great potential for tuning the physical and structural properties by chemical modifications, a panoply of functionalized smart Pluronic^®^ nanovehicles can be designed, and grafting of Pluronic chains with drugs, bioimaging agents, and other functional polymers provides good example of Pluronic^®^ micelles versatility in cancer therapy. The therapeutic efficiency of Pluronic^®^ micelles depends on their ability to overcome biological barriers and deliver and release the therapeutic load at the target site. To resist the shear forces of blood circulation and dissociation of micelles upon dilution in the systemic circulation, micelles can be cross-linked with various stimuli-sensitive linkers. Additionally, sensitivity to various exo- and endo-cellular stimuli can greatly facilitate surpassing of biological barriers and enhance tumor targeting, cellular uptake, and therapeutic cargo release (Figure 6). For example, external stimuli such as hypothermia, radiation, ultrasonic and magnetic irradiation, and photoimmunotherapy can be used to increase tumor perfusion and improve tumor permeability. Although Pluronic^®^ micelles can be internalized by cancer cells *via* clathrin-mediated uptake as observed for P85, selective targeting of tumor tissues and cells has become a routinely used stratagem by decorating the surface of Pluronic^®^ micelles with high-affinity targeting ligands [112]. So far, a wide variety of molecules have been used to induce receptor-mediated endocytosis by targeting receptors that are overexpressed by tumor or endothelial cells [113]. Some Pluronics^®^ can also change the endosomal membrane permeabilization in a cell-dependent manner which results in endosomal escape and drug release [114]. Programmed drug release from Pluronic micelles has also been exploiting stimuli-sensitive linkers that are sensitive to extra- and intra-cellular cues such as pH, redox system, or enzymes. Non-specific electrostatic interactions of Pluronic micelles that were grafted with cationic polymers such as polyethyleneimine with negatively charged cell and organelle membranes can also be exploited to increase cellular uptake but also to deliver negatively charged therapeutics such as nucleic acids, which opened up wide application of Pluronic micelles in combination with drug-gene therapy [115,116,117]. The ability of Pluronic^®^ micelles to be taken up by various immune cells such as myeloid-derived suppressor cells or dendritic cells is increasingly employed in the design of new immunocancer therapies [118,119]. Additionally, some Pluronics^®^ can interact with cell membranes and affect important cellular functions, potentially contributing to the effects of therapeutic load. The attractive inherent antitumor properties of Pluronic^®^ polymers in combination with other functional polymers, variety of therapeutic molecules and cell targeting, and stimuli-responsive ligands greatly improved the antitumoral therapeutic effects of tested drugs and are widely exploited in combination with gene therapy, phothodynamic, or immunotherapy [120,121]. In spite of that, the extraordinary complexity of biological challenges in the delivery of micellar drug payload makes their therapeutic potential still not exploited to the fullest, and therapeutic load stability and off-target delivery still need to be addressed [122,123,124].

### 5.2. Pluronics^®^ for Cancer Theranostics 

Cancer is a complex multistep disease that results from alterations in normal proliferation, differentiation, and cell death mechanisms [125]. Conventional cancer therapy encompasses surgery, radiotherapy, chemotherapy, and/or immunotherapy with poor efficacy and pronounced off-target effects [126]. PMs that are based on Pluronics^®^ are versatile nanoparticles that can be used to treat cancer due to their specific advantages such as biocompatibility, stability, enhanced permeability and retention (EPR) effect, the possibility for precise triggering and targeting decoration, and reduced toxicity [127]. Typically, PMs assemble into a core-shell-like structure, with a hydrophobic inner core appropriate to carrier hydrophobic drugs and a hydrophilic periphery suitable to transporter hydrophilic drugs/cargos for target/triggering the release of drugs or imaging agents [128]. Hence, Pluronic^®^-like PMs are suitable for cancer theranostics with the three-in-one purpose: diagnose, treat, and monitor the disease [102].

Based on this potential, some pre-clinical investigations have been performed using Pluronic^®^ F127 with near-infrared (NIR) fluorescent probes to combine phototherapies and imaging guidance for cancer treatment [129,130,131] (Figure 7).

Pluronics^®^ advantageously present sol-gel transition, making them smart nanosystems [132]. This feature makes these polymers relevant for potential controlled/sustained release of drugs or imaging moieties, contributing to patient compliance and quality-of-life. In this sense, using nanotheranostic Pluronic^®^-like PMs can shed light on the dark shadows of tumors and may represent a precious catapult to bring personalized medicine to clinical reality [102]. 

Despite that, the extraordinary complexity of biological challenges in the delivery of micellar drug payload makes their therapeutic potential still not exploited to the fullest, and therapeutic load stability and off-target delivery still need to be addressed [122,123,124].

## 6. Polymeric Micelles: A Promising Pathway for Dermal Drug Delivery

Nanotechnology is an area that is in great development and application in the most varied fields of science. In cosmetic and pharmaceutical products, conventional formulations for topical application are not always able to effectively penetrate the physical barrier that human skin exerts against factors and compounds of the external environment [133,134,135]. PMs appear as an alternative carrier for drugs and active ingredient delivery, allowing ingredients with lower solubility and higher lipophilicity to be delivered. The augmented bioavailability of drugs, greater efficacy even at a lower dose, and selective drug delivery in specific organelles are exciting advantages of the polymeric micelles usage in the cutaneous application while also reducing many local and systemic adverse effects, as schematized in Figure 8, leading to increased patient compliance to the therapeutics [136,137,138,139].

There are three routes for the active pharmaceutical ingredients to penetrate the skin which are schematized in Figure 9. PMs have the advantage of being able to penetrate by the appendageal route. In diseases affecting the sweat glands and ducts, or the pilosebaceous unit that is composed of the hair shaft, the hair follicle, the sebaceous gland, and the erector pili muscle, the integration of the active ingredient in PMs leads to a targeted delivery of these compounds to these skin organelles. Thus, the correct dose is delivered with lower skin deposition, decreasing the local side effects and augmenting the patient compliance and the efficacy of the treatment [139,140].

Over the last decade, numerous studies have been conducted with formulations containing PMs as drug vehicles for treating skin problems, with safety and efficacy benefits, even at a lower dose, as reviewed by Parra et al. [139] and summarized in Table 3 [139,141,142,143,144,145,146,147,148,149,150].

The characteristics of PMs can be worked to alter their behavior and their efficacy. For example, increasing the length of the hydrophilic fragment, while maintaining the hydrophilic block’s length can influence the micelle’s size and reduce the CMC value. When compared to other nanocarrier systems, PMs-based formulations proved a low toxicity profile in the in vivo trials that were conducted, sometimes so mild that they cannot be qualified as dose-limiting toxicity. Despite these results, the nanotoxicity and econanotoxicity of PMs-based carrier systems must be carefully assessed both in vitro and in vivo, and further guidance and regulation of the tests should be developed by the regulatory agencies [139,144,151,152].

## 7. Micelleplexes: The Key to Achieving Success in Therapy

Recently, a synergistic association between genetics and nanotechnology has emerged to improve therapeutics, particularly in the field of cancer [153,154,155,156]. This junction allowed the concept of different therapeutic weapons from the production of nanosystems that are capable of efficiently transporting ribonucleic acid (RNA) or deoxyribonucleic acid (DNA) as well as drugs [157]. One of the most promising non-viral vectors are micelleplexes [158,159,160,161]. These nanosystems are composed of amphiphilic copolymers, a cationic polymer, genetic material, and a drug [162]. The micelleplexes have the ability to spontaneously organize themselves to form PMs with cationic properties, with a size between 10 and 100 nm. These PMs are molecules that increase the stability of the encapsulated genetic material and allow its passage through physiological membranes, and, for these reasons, can be successfully used in therapy. Furthermore, due to its structural organization, the micelleplex directs the drug more efficiently, it protects the genetic material from the action of the nucleases, and, as it presents a positive charge, will promote electrostatic interactions with the membrane, facilitating the internalization of the complex in the cells by endocytosis (Figure 10) [157,163,164,165].

Currently, there are several studies and trials involving the use of micelleplexes, demonstrating themselves as potential candidates for the treatment of cancer (Figure 10). In a recent study, researchers at the University of Michigan, proved the ability of a folate-decorated micelleplex to deliver a targeted dose of siRNA and paclitaxel in order to overcome chemotherapy resistance, which can frequently cause complications in ovarian cancer patients. The siRNA that was incorporated into the micelleplex aimed to silence the expression of toll-like 4 receptors (TLR-4), which are responsible for the proliferation of tumor cells and the inhibition of apoptosis. Their results revealed that the formulated micellexes are able to decrease the overexpression of TLR-4, and the resistance of SKOV-3 cells to the conventional therapy with paclitaxel, together with reduced toxicity in healthy cells [166]. Another study demonstrated the potential of a micelleplex in RNA delivery. In this study, the polymer N-succinyl-chitosan poly(lysine) palmitic acid was used. N-succinyl chitosan was used to increase the micelle half-life and decrease the toxicity of poly(lysine). In turn, poly(lysine) constituted the cationic part, electrostatically condensing the negatively charged siRNA. Palmitic acid constituted the hydrophobic core where doxorubicin (Dox) was incorporated. It was concluded that this micelleplex can improve the therapeutic efficacy in vitro and in vivo, overcoming the efflux of drug out of the tumor cell, by decreasing the gene expression of P-glycoprotein and consequently promoting an increase in the concentration of Dox inside the tumor. This study showed the effectiveness of Dox–siRNA-micelle for tumor-targeted delivery, multidrug resistance reversal, and antitumor activity, and provided an effective strategy for the treatment of cancers that develop multidrug resistance [167]. In another study performed by Qian et al. [168], they formulated micelleplexes for the treatment of glioma using a star-branched amphiphilic copolymer incorporating a microRNA-21 inhibitor in combination with Dox. The amphiphilic copolymer was composed of a hydrophobic core consisting of PLA to incorporate the drug and a hydrophilic periphery comprised of Poly[2 (Dimethylamino)ethyl Methacrylate] (PDMAEMA) that confers high buffering capacity, low toxicity, and promotes renal clearance. The developed micelleplexes significantly reduced the expression of the anti-apoptotic protein B-cell lymphoma 2 (BCL-2) and decreased the tumor volume. These results may indicate that star-branched amphiphilic copolymer is promising in the delivery of hydrophobic drugs and genes [168].

## 8. Polymeric Micelles Limitations and Their Respective Solutions

Although there are numerous advantages to the use of PMs as drug delivery nanosystems, it is important to evaluate their possible limitations. Sometimes, the PMs could present certain limitations such as toxicity, such as in HepG2 cells [140], and lack of stability in blood. For these reasons, investigators have developed several methods to overcome the adverse effects that are associated with the use of PMs. Table 4 provides a general overview of the various strategies that can be used to minimize the possible limitations that are associated with PMs.

## 9. Conclusions and Future Perspectives

PMs are versatile alternative drug nanocarriers with utmost properties compared to other micellar systems [196,199,200,201]. They are endowed with the incorporation of considerably higher levels of drugs, increased blood circulation time, and thermodynamic stability. Engineering of the PMs core leads to maximum drug loading capacity and blood circulation half-life. The length of hydrophobic blocks and the nature of substituents that are present in the core mainly control the drug loading capacity of PMs [202]. Pluronics^®^ are the most used and studied polymers to develop PMs. Pluronics^®^ are U.S. Food and Drug Administration-approved, which are widely used for solubilization and delivery of drugs, gene therapy, diagnostics, and tissue engineering applications, making it an extremely relevant nanosystem to consider for drug transport. Related to micelleplexes, numerous studies have demonstrated several advantages, namely by their ability to cross cancer cell membranes, biodistribution, low toxicity, biodegradability, increase the stability of the encapsulated genetic material, as well as high efficiency in the transport and delivery of drug and genetic material to target cells. However, PMs also present some drawbacks, e.g., immunogenicity and instability at a low pH, that may hampers their translation into clinical practice [203]. Here, it was revisited some methods that can improve the stability and reduce the immunogenicity of PMs, namely by the inclusion of PEG chains, that alters the pharmacokinetic profile, providing a more favourable clearance and tissue distribution. Additionally, POx-based PMs have revealed potential benefits in cancer treatment due to their unique characteristics. They significantly enhance the solubility and stability of drugs and improve their efficacy and safety in a transformative way. The development and validation of in silico strategies to enable the delivery of poorly soluble drugs and established critical physicochemical parameters to maximize drug loading, formulation stability, and tumor exposure have demonstrated promising features. Models predicting drug loading efficiency and loading capacity using novel descriptors of drug-polymer complexes were employed for the virtual screening of drug libraries. Using a rational computer-aided approach, POx micelles that were loaded with chemosensitizers, anticancer agents, and immunomodulating small molecules were designed to improve the treatment of non-small cell lung cancer. The POx PMs approach has shown significant promise for the delivery and co-delivery of agents that target the tumor microenvironment as a strategy for cancer immunotherapy. Usually, complete formulations containing PMs are too complex to be efficiently tackled computationally by molecular simulation. The molecular weight of the components, and aspects pertaining to the respective concentration make it difficult to model these systems. However, the information that is acquired from modelling a part of the overall system, focusing on some types of interactions is usually very informative and can be easily extended to other systems. Also, increasing computational power means increasing the model size and extended simulation time, for instance, when one is using molecular dynamics, which means more relevant information can be extracted. It is thus predicted that computer simulation will be more often used for “virtual screening” or rationalization of multi-component formulations.

Another interesting applicability of PMs is in dermal drug delivery. Several studies have revealed the therapeutic applications of PMs in acne vulgaris, psoriasis diseases, dermal fungal infections, and anti-ageing products. All of these studies demonstrated that using PMs as drug loading nanosystems can benefit the safety and efficacy of the treatment, even at a lower dose.

Despite the barriers and the “death valley” that may hamper the translation of more PMs into clinical reality, there is ample evidence that PMs are potential candidates for treating contemporary diseases. 

Overall, PMs can constitute promising nanosystems for the transport of drugs, which allows the establishment of versatile functions based on innovative nanobiotechnology for the development of advanced therapies aiming to overcome failures in conventional treatments and to achieve successful therapies for incurable diseases in the near future.

## Figures and Tables

**Figure 1 pharmaceutics-14-01700-f001:**
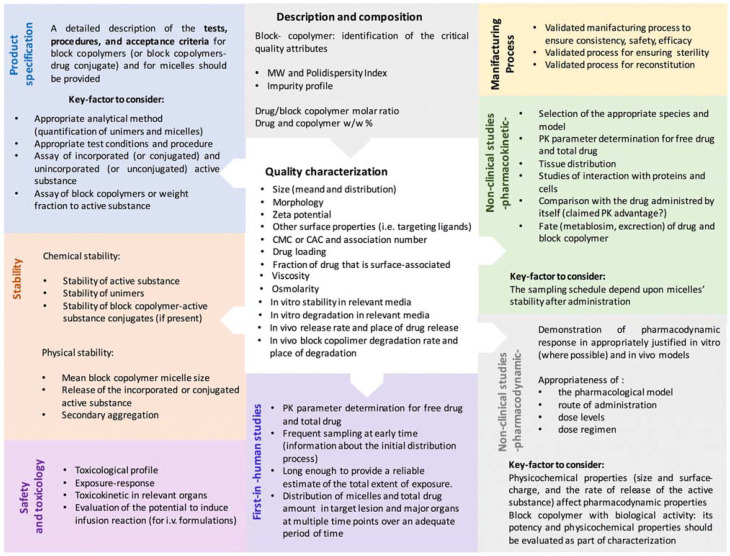
Schematic illustration of the main topics addressed in the Reflection Paper of the European Medicines Agency (EMA) regarding the “Development of block-copolymer-micelle medicinal products”. Reprinted from [16] under a CC-BY 4.0 license.

**Figure 2 pharmaceutics-14-01700-f002:**
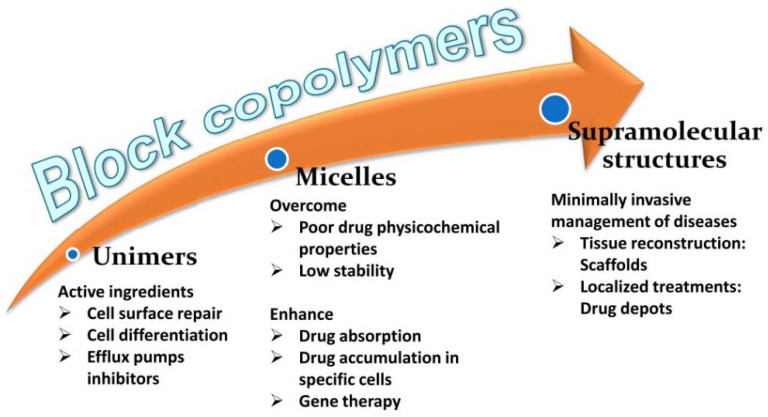
The evolution of the structure of block copolymers. Some advantages of the use of polymeric micelles and supramolecular structures.

**Figure 3 pharmaceutics-14-01700-f003:**
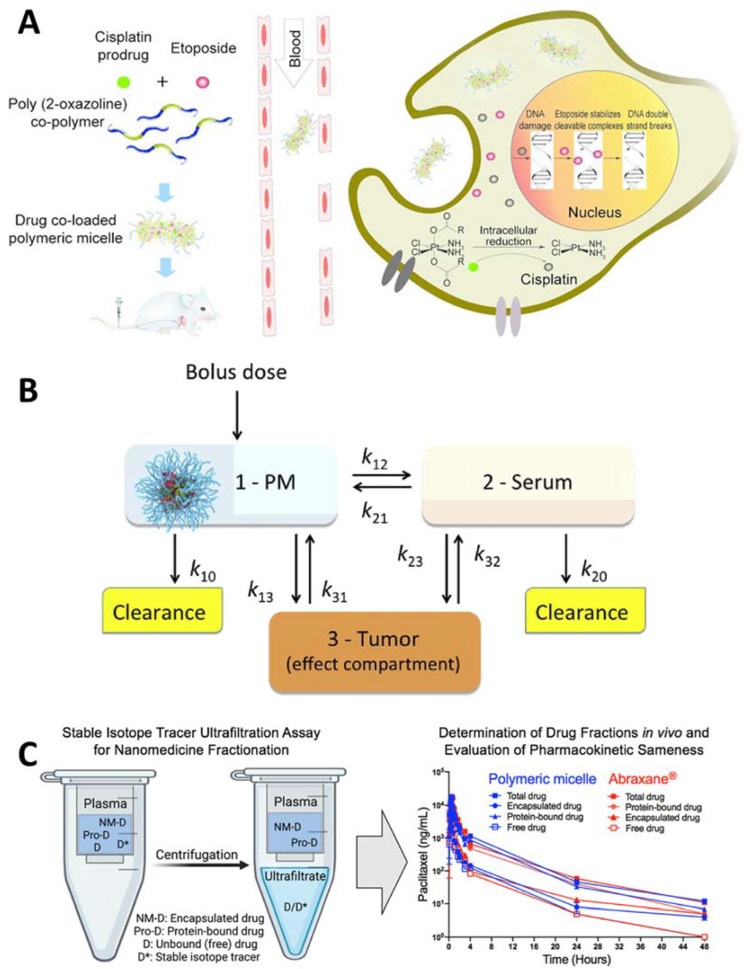
Examples of the application of Poly(2-oxazoline) (POx)-based polymeric micelles and the assessment of their pharmacokinetics (PK) and bioequivalence profiles. In (**A**) poly(2-methyl-2-oxazoline-*block*-2-butyl-2-oxazoline-*block*-2-methyl-2-oxazoline) (P(MeOx-*b*-BuOx-*b*-MeOx) is used with an alkylated cisplatin prodrug to enable co-formulation of etoposide (ETO) and platinum drug combination (“EP/PE”) in a single high-capacity vehicle to improve the treatment of small cell lung cancer (SCLC). The drugs co-loading in the micelles result in a slowed-down release, improved pharmacokinetics, and increased tumor distribution of both drugs. Reprinted with permission from [58]. Copyright 2018 American Chemical Society. In (**B**) a three-compartmental model describing the PMs drug delivery to a tumor. The drug is administered as bolus in the form of PMs (1) and is subsequently distributed between the serum (2) and tumor (3) compartments. The PK constants correspond to: *k*_12_—rate of drug transfer from PMs to serum; *k*_21_—rate of drug re-capture from serum to PMs; *k*_13_—rate of transfer (permeability) of the micellar drug to tumor; *k*_23_—rate of transfer of the serum-bound drug to tumor; *k*_31_ and *k*_32_—rates of drug reabsorption from tumor to PMs and serum, respectively; and *k*_10_ and *k*_20_—micellar and serum-bound drug clearances, respectively. The model assumes that the drug solubility in blood is very low and the free drug form in the blood is, therefore, neglected. Reprinted from [59], Copyright 2018, with permission from Elsevier. In (**C**) a comprehensive preclinical assessment of the poly (2-oxazoline)-based polymeric micelle of paclitaxel (PTX) (POXOL hl-PMs), including bioequivalence comparison to the clinically-approved paclitaxel nanomedicine, Abraxane^®^. Reprinted from [67], Copyright 2018, with permission from Elsevier.

**Figure 4 pharmaceutics-14-01700-f004:**
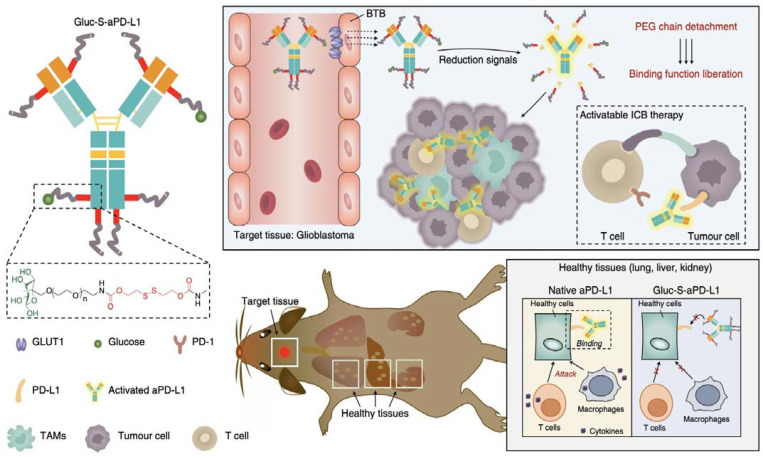
Glycosylated-PEGylation of anti-PD-L1 antibodies for creating a targeted immunotherapy against glioblastoma. The glycosylated-PEGylation allowed the active transport of the antibody through the BBTB of glioblastoma by targeting the GLUT1 on the endothelial cells. Inside the tumor, the antibodies were dePEGylated by the cleavage of the reduction-sensitive bonds. In off-target tissues, the coated antibodies remained silent, avoiding the immune-related adverse events. Reprinted with permission from reference [76]. Copyright Nature 2021.

**Figure 5 pharmaceutics-14-01700-f005:**
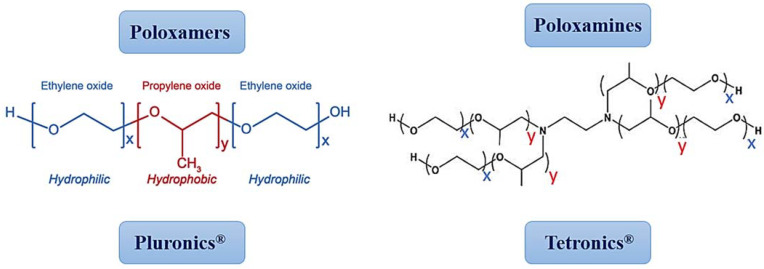
Structures of poloxamers/Pluronics^®^ and poloxamines/Tetronics^®^.

**Figure 6 pharmaceutics-14-01700-f006:**
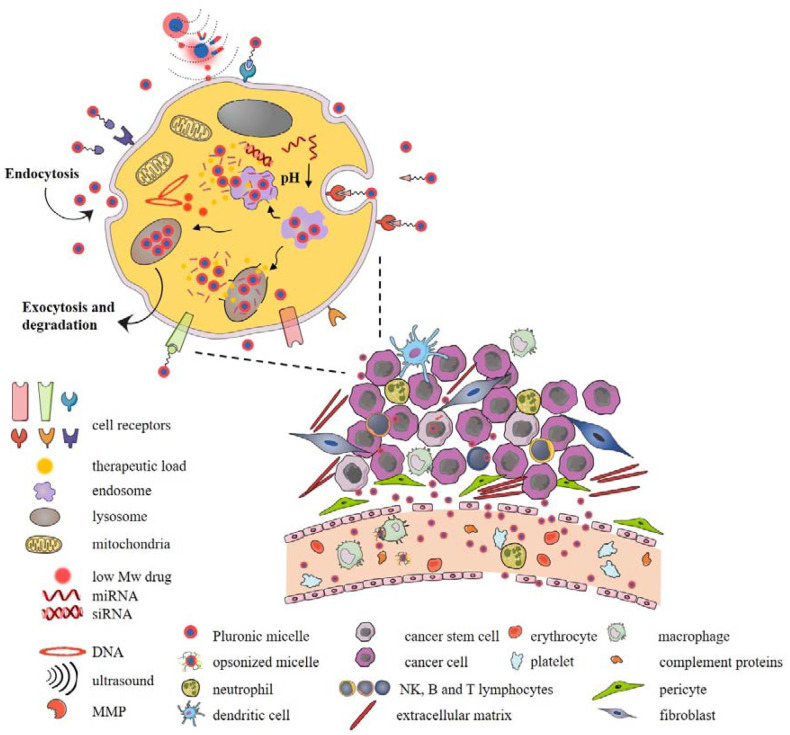
Routes of internalization and interactions of Pluronic^®^-like polymeric micelles and cancer cells.

**Figure 7 pharmaceutics-14-01700-f007:**
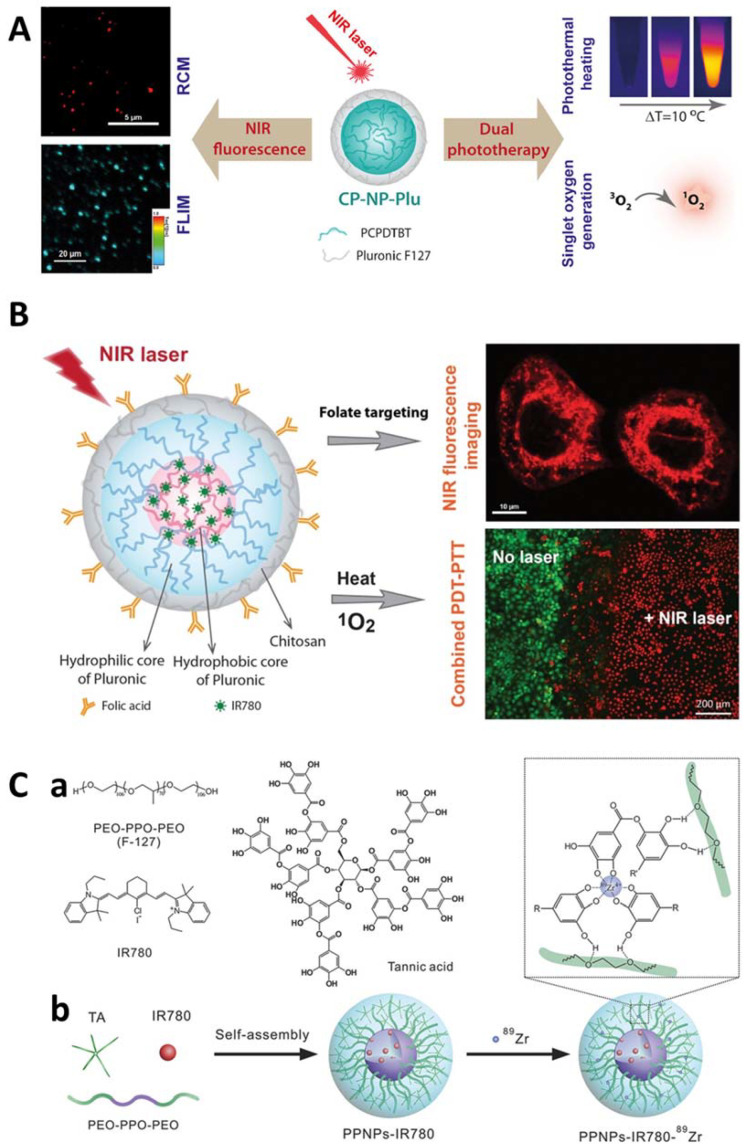
Examples of Pluronic^®^-based approaches for nanotheranostics. In (**A**), it schematically represents the development of Pluronic^®^ F127 stabilized conjugated polymer nanoparticles for Near Infra-Red (NIR) fluorescence imaging and dual phototherapy applications. Reprinted from [130] Copyright 2021, with permission from Elsevier. In (**B**), is a representative scheme of the formulation of folate-targeted Pluronic^®^ F127-chitosan nanocapsules that are loaded with Infra-Red (IR)780 for near-infrared fluorescence imaging and photothermal-photodynamic therapy of ovarian cancer. Reprinted from [131] Copyright 2021, with permission from Elsevier. In (**C**), it represents a schematic overview of the polyphenol–Pluronic^®^ self-assembled supramolecular nanoparticles (PPNPs) for tumor NIR fluorescence/Positron Electron Transmission (PET) imaging. Particularly, in (**C**, **a**) it represents the chemical structure of Pluronic^®^ F127, TA and IR780. In (**C**, **b**) a schematic illustration of the synthesis of PPNPs-IR780-^89^Zr. The polyphenols in Tannic Acid (TA) have strong hydrogen bonding with the PEO chain in F127. IR780 was loaded by hydrophobic interaction with Polypropylene oxide (PPO) chain in F127. ^89^Zr was chelated by excess phenol groups in TA. Reprinted from [129] Copyright 2021, with permission from Elsevier.

**Figure 8 pharmaceutics-14-01700-f008:**
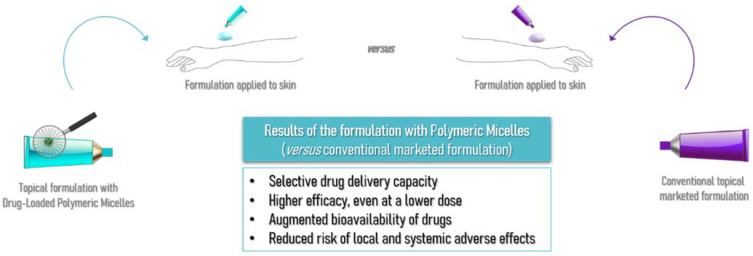
Schematic representation of the comparison between conventional topical marketed formulations and formulations containing polymeric micelles as drug carriers, and the results that were obtained.

**Figure 9 pharmaceutics-14-01700-f009:**
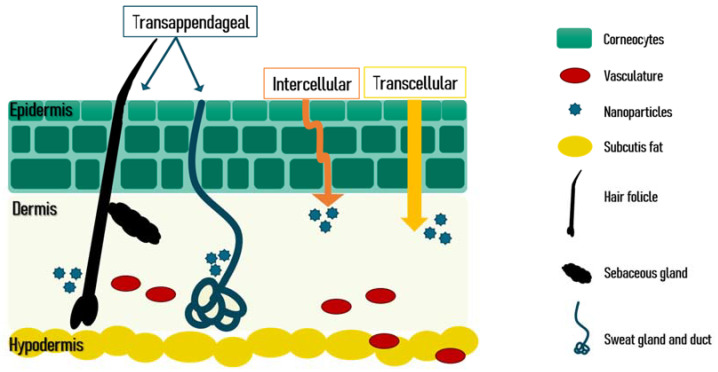
Schematic view of the skin layers and skin penetration routes. Reprinted from [139] under a CC BY license.

**Figure 10 pharmaceutics-14-01700-f010:**
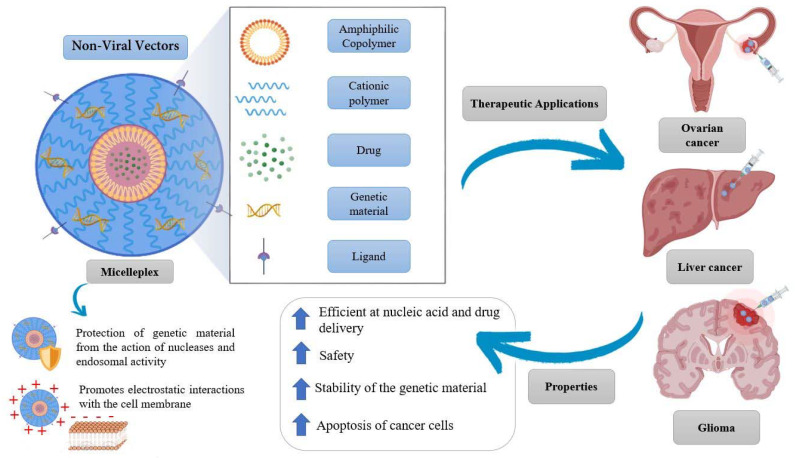
Basic structure and summary of the properties of micelleplexes as well as their therapeutic application in ovarian cancer, liver cancer, and glioma.

**Table 1 pharmaceutics-14-01700-t001:** Examples and respective properties of hydrophobic and hydrophilic polymers as well as amphiphilic block copolymers commonly used in the formulation of polymeric micelles.

	Polymers	Properties	References
**Hydrophobic Polymers**	poly(D,L-lactide)–PLA	PLA-based PMs are clinically-approved (Genexol^®^, Nanoxel^®^).	[11,12,13,14,15,17,18]
	poly(lactic-co-glycolic acid)–PLGA	PLGA is used as a biodegradable surgical suture in the clinic (Vicryl^®^). Biodegradable.	
	poly(β-benzyl-l-aspartate)	The presence of the benzyl group grants increased hydrophobicity. Biodegradable.	
	poly(γ-benzyl-α, l-glutamate)	The presence of the benzyl group grants increased hydrophobicity. Ultra-high loading capacity for various poorly soluble drugs (ex. paclitaxel, etoposide) as well as a versatile library of polymer structures.	
	poly(2-n-butyl-2-oxazoline)	The presence of the benzyl group grants increased hydrophobicity. Ultra-high loading capacity for several poorly soluble drugs, such as curcumin.	
**Hydrophilic Polymers**	polyethylene glycol (PEG)	Has been used in clinically-approved nanoformulations including PMs (Genexol^®^ PM).	
	poly(2-methyl-2-oxazoline)–PMeOx	PMeOx is more hydrophilic than PEG.	
	poly(sarocosine)	Evaluated as PEG replacement. Biodegradable.	
	dextran	Has been used as a component in block and graft copolymers. Has highly variable molecular weight and dextran has been used as an excipient in clinically-approved injectable products (Feraheme^®^). Biodegradable.	
**Amphiphilic block copolymers**	poly(propylene oxide)–PPO poly(ethylene oxide)–PEO PEO_n_-PPO_m_-PEO_n_	PEO_n_-PPO_m_-PEO_n_ triblock copolymers are usually used in pharmaceutical formulations as non-active pharmaceutical ingredients. Pluronic^®^-based PMs entrapping Doxorubicin, SP1049C, had entered clinical trials and have been granted orphan drug designation by the FDA. Commercially available as poloxamers (Pluronic^®^). Biocompatible.	

**Table 2 pharmaceutics-14-01700-t002:** Summary of the properties of the various nanosystems as well as their advantages and limitations.

Nanosystem	Size	Advantage	Limitations	References
Solid lipid nanoparticles (SLN)	50–1000 nm	Biocompatible; Biodegradable; High drug loading; Good stability; Enhanced bioavailability; Excellent nanocarriers for controlled release and for targeted drug delivery to the reticuloendothelial system.	Costly and complex methods of preparation; Expulsion of the drug from the SLNs over time; Only suitable for loading hydrophobic drugs.	[16,30,36,37,38]
Liposomes	25–2500 nm	Loading simultaneously with two drugs (hydrophobic and hydrophilic); Easy functionalization of the surface; Biocompatible; Low toxicity; Biodegradable.	Costly and complex methods of preparation.	
Nanoemulsions	<100 nm	Loading simultaneously with two drugs (hydrophobic and hydrophilic); Facilitate the bioenhancement of hydrophobic drugs;	Not form spontaneously; Considerable energy is required to generate nanoemulsions; Limited stability; Lack of controlled release functions; Tendency to flocculate and coalescent.	
Micelles	5–100 nm	Easy loading of hydrophobic drug; Enhanced permeability; Low toxicity; Extended blood half-life	Low loading efficacy; Instability.	
Polymeric nanoparticles	<1000 nm	High drug loading capacity; Drug release regulated by selecting; Appropriate preparation methods; High stability; High membrane permeability; Biodegradable; Easy functionalization of the surface.	Costly and complex methods of preparation; Prone to aggregation and opsonization in the bloodstream; Need of functionalization.	
Polymeric micelles	10–100 nm	Easy and high loading hydrophobic drug; Drug release regulated by polymers structure; Small size; Prevention of rapid clearance by reticuloendothelial system; Low CMC Easy and cheap preparation; Biocompatible; Extended circulation time; Lower toxicity of a drug; High stability in vitro and in vivo	Complex characterization; Lack of stability in blood; Limited number of polymers for use; Lack of suitable methods for scale-up; Dependency of critical micelle concentration.	
Dendrimers	1–10 nm	High drug loading capacity; Small size; Versatility of surface functionalization.	High cytotoxicity; Haemolytic properties; Non-biodegradable; Not a good candidate carrier for hydrophilic drugs; Elimination and metabolism depending on the generation and materials; High cost for their synthesis.	
Inorganic Nanoparticle	1–100 nm	Stimuli-responsive behavior; Good microbial resistance and good storage properties; Versatility of surface functionalization.	Poor data regarding long-term exposure; Toxicity and instability.	
Nanocrystal	<500 nm	Well-understood and established manufacturing techniques; Excellent reproducibility; Applicable to drugs with different solubility profiles; Suitable for oral administration;	Requires high energy input that drives up costs; Needs further modification to ensure stability; Lack of controlled release functions.	[39]

**Table 3 pharmaceutics-14-01700-t003:** Summary of the polymeric micelles used for skin delivery described in the literature during the past decade. Reprinted from [139] under a CC BY license.

Active Compounds	Polymers Used in the Composition of Micellar Carrier	Conclusions	Ref.
Anti-Ageing			
Oleanolic Acid	Poloxamer 407	Enhancement in the efficacy of wrinkle alleviation treatment	[141]
Hyaluronan	Oleyl-hyaluronan Hexyl-hyaluronan	Drug reached deeper layers in porcine skin	[143]
CoQ10	Oleyl-hyaluronan	Enhancement in skin hydration	[143]
Acne Vulgaris			
All-trans Retinoic Acid (Tretinoin)	Poly(ethylene glycol)-conjugated Phosphatidylethanolamine	Higher stability profile with slower drug oxidation	[144]
All-trans Retinoic Acid (Tretinoin)	Diblock methoxy-poly(ethylene glycol)-poly(hexyl-substituted lactic acid)	Higher efficiency than marketed formulations	[144]
Adapalene	D-α-tocopheryl polyethylene glycol 1000 succinate	Targeted drug delivery capacity Higher efficiency at lower dose than the marketed formulations	[145]
Benzoyl Peroxide	Pluronic^®^ F127	Targeted drug delivery capacity	[146]
Psoriasis			
Tacrolimus	Diblock Methoxy-poly(ethylene glycol)-poly(hexyl-substituted lactic acid)	Enhancement in skin drug deposition	[147]
Resveratrol	Pluronic^®^ P123 Pluronic^®^ F127	Decrease in the cytokine levels	[148]
Silibinin	-	Reduction of psoriasis index area	[149]
Fungal Infections			
Clotrimazole Econazole nitrate Fluconazole	Methoxy-poly(ethylene glycol)-poly(hexyl-substituted lactic acid)	Enhancement in skin drug deposition	[150]
Terconazole	Pluronic^®^ P123 Pluronic^®^ F127 Cremophor EL	Higher permeation Higher skin deposition	[142]

**Table 4 pharmaceutics-14-01700-t004:** Strategies to overcome the limitations related to the application of polymeric micelles.

Limitations	Strategies	References
Toxicity and Immunogenicity	PEGylation approach; Use pH-sensitive micelles; High affinity targeting ligands; Use biodegradable and biocompatible PMs.	[169,170,171,172,173,174,175]
Low Stability	PEGylation approach; Covalent cross-linking strategies: Shell cross-linked micelles, core cross-linked micelles. Covalent cross-linking methods: Photo/ultraviolet-induced dimerization, di-functional cross-linkers, click cross-linking method, silicon chemistry method, reversible boronate ester bond; Non-covalent cross-linking strategies: Complexation of micelle cores, macrocyclic host-guest complexation; Altering hydrophilic/hydrophobic block ratios of the micelles; Increase of the crystallinity of hydrophobic segments; Introduction of inorganic materials into the core or shell of micelles to act as structural stabilizers.	[176,177,178,179,180,181,182,183,184,185,186,187]
Non-biodegradable and non-biocompatible	Use biodegradable PMs such as: poly(ethylene glycol) (PEG), polylactic acid (PLA), poly(caprolactone) (PCL), polyglycolic acid (PGA), monomethoxy poly (ethylene glycol)-block-poly(D,L-lactide) (mPEG-PDLLA), poly(L-histidine), poly(L-lactic acid) (PLLA), PEG-poly(S-(2-nitrobenzyl)-l-cysteine), phospholipid, such as 1,2-distearoyl-sn-glycero-3-phosphoethanolamine (DSPE).	[188,189,190]
Low drug loading	Improving the compatibility between drug and polymer; Polymeric prodrugs; Electrostatic interactions; Cross-linking of the core or the shell of self-assembled PMs; Layer by layer coating of PMs; Host-guest complex micelles; Micelle-like nanoparticles; Integrate drug attached polymers into lipids.	[191,192,193,194,195]
High CMC	Increasing chain length of the hydrophobic block; Decoration of micelle cores with various fatty acid; Addition of benzyl groups.	[179]
Rapid clearance	PEGylation approach; Cross-linked with various stimuli-sensitive linkers.	[75,112]
Low selectivity	PEGylation approach; High-affinity targeting ligands.	[75]
Low membrane disrupting capability	Hydrophobic moieties and cationic groups; Polymers with buffering capacity at endosomal pH; High-affinity targeting ligands.	[89,95,196]
Low efficiency in drug delivery	Cross-linked with various stimuli-sensitive linkers; High-affinity targeting ligands; Intracellular redox-responsive drug release.	[112,197,198]

## Data Availability

Not applicable.

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
