# Peer review of "New Advances in Biomedical Application of Polymeric Micelles"

_pharmaceutics, 2022, doi:10.3390/pharmaceutics14081700_

Round 1
Reviewer 1 Report
The review article entitled “New Advances in Polymeric Micelles for Therapeutic Applications” is an interesting article and the topic is fashion. Unfortunately, I cannot suggest publishing this article at the current state. I gained the impression that this review article was submitted in an unfinished condition! Some serious questions and issues should be addressed. Please see the following points:
1. The keywords seems to me not suitable. “New advances”? This is too general. Copolymers should be added.
2. “Therapeutic applications” – I cannot cope with this term in the keywords and the title and I would not search for that keyword. Maybe “biomedical applications” would be more suitable? This is no request to change it, it is just a suggestion.
3. Space sign are missing. Check the article again.
4. I have the impression that this review article is similar to this very good review article: https://doi.org/10.1016/j.jconrel.2021.02.031 - The authors should explain why their article is different and what is new compared to that in the controlled release journal. Until now, I cannot gain much new knowledge for what a new review article in this field could be useful.
5. In the introduction should be strengthen what is the difference between a PM and a common surfactant. Please, also explain is the CMC also relevant to PMs.
6. In the introduction should also be discussed the differences of PMs in comparison to nanoemulsions.
7. The figure labelling should be checked. The Figure 1 is used for the first and second figure. Therefore, the whole figure labeling must be checked and fixed.
8. Copolymers are important for PMs. Therefore, a suitable overview of the commonly used copolymers should be added to the manuscript.
9. For the formations of PMs/micelles from copolymers, the Ostwald ripening plays a rule. This is point is completely missing, especially for chapter 2.
10. Chapter 9, conclusion and future perspectives is too short. Please extend it. Especially the different fields of medicine are described too general. Please give more concrete perspectives for different fields of medicine (skin, cancer treatment in the body, …)
Author Response
Manuscript ID: pharmaceutics-1772803
Title: New Advances in Polymeric Micelles for Therapeutic Applications
Dear Editor,
Doctor Yoona Xu
The authors appreciate the careful review and greatly acknowledge the comments that the Reviewer has provided on our manuscript. We have carefully revised the manuscript and have made the recommended changes and answered in detail to the questions raised. Additional information was also added when appropriate. All changes made to the text are highlighted in green colour and marked up using the “Track Changes” function.
# Reviewer 1
The review article entitled “New Advances in Polymeric Micelles for Therapeutic Applications” is an interesting article and the topic is fashion. Unfortunately, I cannot suggest publishing this article at the current state. I gained the impression that this review article was submitted in an unfinished condition! Some serious questions and issues should be addressed. Please see the following points:
- The keywords seems to me not suitable. “New advances”? This is too general. Copolymers should be added.
Response: The authors thank for the comment. The keywords were altered to better suit the content of the manuscript. Please check lines 48 and 49, in turquoise.
- “Therapeutic applications” – I cannot cope with this term in the keywords and the title and I would not search for that keyword. Maybe “biomedical applications” would be more suitable? This is no request to change it, it is just a suggestion.
Response: The authors thank for the suggestion which was adopted. The title and keywords were adapted accordingly.
- Space sign are missing. Check the article again.
Response: The manuscript was thoroughly revised in order to include the missing spaces.
- I have the impression that this review article is similar to this very good review article: https://doi.org/10.1016/j.jconrel.2021.02.031 - The authors should explain why their article is different and what is new compared to that in the controlled release journal. Until now, I cannot gain much new knowledge for what a new review article in this field could be useful.
Response: The authors acknowledged the referred article. It was very insightful and added to the references. After thorough analysis of the article, the main differences reside in: 1. The reference to in silico to predict nanoparticles properties, such as physicochemical properties, drug loading maximization, and stability; 2. Referral to the most recent studies in cancer and dermatological applications; 3. A thorough comparison of copolymers and nanosystems; 4. Limitations of the nanosystems as nanocarriers and possible solution; 5. Summary of the EMA reflection paper on polymeric micelles.
- In the introduction should be strengthen what is the difference between a PM and a common surfactant. Please, also explain is the CMC also relevant to PMs.
Response: The difference between PM and common surfactants as well as the relevance of CMC to PMs were developed, accordingly. Please, check lines 82 and following.
- In the introduction should also be discussed the differences of PMs in comparison to nanoemulsions.
Response: The authors are grateful for this suggestion. This was presented in Table 2 (line 111).
- The figure labelling should be checked. The Figure 1 is used for the first and second figure. Therefore, the whole figure labeling must be checked and fixed.
Response: The figure numbering was revised throughout the manuscript. The authors appreciate the noticing of this lapse.
- Copolymers are important for PMs. Therefore, a suitable overview of the commonly used copolymers should be added to the manuscript.
Response: A comparative overview of copolymers for PMs was included as Table 1. Please see line 79.
- For the formations of PMs/micelles from copolymers, the Ostwald ripening plays a rule. This is point is completely missing, especially for chapter 2.
Response: The authors are grateful for this suggestion The Ostwald ripening was added to the manuscript. Please check lines 86 and following.
- Chapter 9, conclusion and future perspectives is too short. Please extend it. Especially the different fields of medicine are described too general. Please give more concrete perspectives for different fields of medicine (skin, cancer treatment in the body, …)
Response: The authors appreciated the reviewer comment. The conclusion and future perspectives were extended and more deeply discussed accordingly. Please vide lines 416 and following.
Reviewer 2 Report
Authors summarized advanced in polymeric micelles based on reports presented in “1st Advances in Polymeric Micelles for therapeutic applications". The overall aim of the review is justified and addressed an important topic in pharmaceutics. I have following suggestions to improve the manuscript;
1. In title "New advances in therapeutic applications of polymeric micelles" is suggested.
2. Abstract should be revised carefully to describe the content of review. Present abstract is not well written to describe the review content and seems more like introduction section. Many lines required reference to support like "PMs are difficult to recognize by the macrophage phagocytic system; present high circulation half-time; improve the pharmacokinetic profile of the encapsulated drug; present high load capacity; and are synthesized by a reproducible, easy, and low-cost"
3.Introduction should be elaborate. I suggest to add abstract content in introduction and revise the abstract.
4. line 72 renamed should be revised as renowned.
5. Main content: Section headings should be revised like 5 and 6 sections may be combined because both are focused on Pluronic®-based PMs.
6. Addition of a section of about role of PM in enhancement of oral bioavailability of poorly soluble drugs will be helpful for readers.
7. Line 377-385. Authors should enlist all possible limitations about PMs. The authors input about key steps to solve these limitation in detail should be added. I suggest to add a Table about limitations and proposed solutions.
Author Response
Manuscript ID: pharmaceutics-1772803
Title: New Advances in Polymeric Micelles for Therapeutic Applications
Dear Editor,
Doctor Yoona Xu
The authors appreciate the careful review and greatly acknowledge the comments that the Reviewer has provided on our manuscript. We have carefully revised the manuscript and have made the recommended changes and answered in detail to the questions raised. Additional information was also added when appropriate. All changes made to the text are highlighted in blue colour and marked up using the “Track Changes” function.
# Reviewer 2
Authors summarized advanced in polymeric micelles based on reports presented in “1st Advances in Polymeric Micelles for therapeutic applications". The overall aim of the review is justified and addressed an important topic in pharmaceutics. I have following suggestions to improve the manuscript;
- In title "New advances in therapeutic applications of polymeric micelles" is suggested.
Response: The authors thank for the suggestion which was adopted.
- Abstract should be revised carefully to describe the content of review. Present abstract is not well written to describe the review content and seems more like introduction section. Many lines required reference to support like "PMs are difficult to recognize by the macrophage phagocytic system; present high circulation half-time; improve the pharmacokinetic profile of the encapsulated drug; present high load capacity; and are synthesized by a reproducible, easy, and low-cost"
Response: The authors thank for the comment. The abstract was altered to better suit the content of the manuscript. Please check lines 22 and following. The respective references were added.
3.Introduction should be elaborate. I suggest to add abstract content in introduction and revise the abstract.
Response: The introduction was extended, accordingly and the abstract was revised. Please check lines 53 and following.
- line 72 renamed should be revised as renowned.
Response: The authors thank for the comment. The lapse was corrected. Please check line 116.
- Main content: Section headings should be revised like 5 and 6 sections may be combined because both are focused on Pluronic®-based PMs.
Response: The authors appreciate the comment. The sections were revised and especially sections 5 and 6 were combined.
- Addition of a section of about role of PM in enhancement of oral bioavailability of poorly soluble drugs will be helpful for readers.
Response: The authors thank the suggestion. This content was added to introduction. Please check lines 64 and following.
- Line 377-385. Authors should enlist all possible limitations about PMs. The authors input about key steps to solve these limitation in detail should be added. I suggest to add a Table about limitations and proposed solutions.
Response: The authors thank for the comment. A table has been added. Please check table 4. (lines 406 and following).
Reviewer 3 Report
This manuscript review and introduce Advances in nanomedicine over the past decade. The author confirmed the usefulness and development potential of polymeric micelles by reviewing papers on polymeric micelles within the last 10 years. So, it is suitable for publication in the journal "Pharmaceutics" However, it has the following revised parts. They should be checked prior to the publication. Followings are recommended for the revision.
Minor revision
1. In Figure 2 and Figure 4, it seems that it is necessary to put a caption.
Major revisions
1. The literature is need citations from recent(2012-2022). Some quite outdated papers have been cited, so this needs correction.
2. This paper need around 200 reference for this topic as there are so many papers talking about the polymeric micelle for therapeutic application.
Author Response
Manuscript ID: pharmaceutics-1772803
Title: New Advances in Polymeric Micelles for Therapeutic Applications
Dear Editor,
Doctor Yoona Xu
The authors appreciate the careful review and greatly acknowledge the comments that the Reviewer has provided on our manuscript. We have carefully revised the manuscript and have made the recommended changes and answered in detail to the questions raised. Additional information was also added when appropriate. All changes made to the text are highlighted in red colour and marked up using the “Track Changes” function.
# Reviewer 3
This manuscript review and introduce Advances in nanomedicine over the past decade. The author confirmed the usefulness and development potential of polymeric micelles by reviewing papers on polymeric micelles within the last 10 years. So, it is suitable for publication in the journal "Pharmaceutics" However, it has the following revised parts. They should be checked prior to the publication. Followings are recommended for the revision.
Minor revision
- In Figure 2 and Figure 4, it seems that it is necessary to put a caption.
Response: Thank you for your comment. A caption was added to Figure 2 and 4.
Major revisions
- The literature is need citations from recent (2012-2022). Some quite outdated papers have been cited, so this needs correction.
Response: The authors thank for the helpful remark. More recent references were added and older ones removed when deemed fit.
- This paper need around 200 reference for this topic as there are so many papers talking about the polymeric micelle for therapeutic application.
Response: The reference list and respective content was updated and extended in order to comprise the more recent advances in the field.
Reviewer 4 Report
This review article summarizing the new advances and therapeutic applications of polymeric miclles is a very good work that deserves publication after some improvement:
1- In page 8: the structure of the Pluronics should be corrected as they are are amphiphilic tri-block copolymers [Polyethylenoxide(PEO)a- Polypropylenoxyde(PEO)b- Polyethylenoxide(PEO)a]; with the letter "y" not "i".
2- Also the authors should mention that the Pluronics carry another name which is poloxamers that should be mentioned in the text.
3- In section 6 dealing with the use of pluronics in cancer therapy, the authors should refer to: 10.1016/j.molliq.2015.12.007, as it mentions the role of pluronics in increasing the cytotoxicity against HEP-G2 cells.
4- Section 7 needs more examples for the use of polymeric micelles in topical and transdermal delivery and the suggested penetration pathways of these polymeric carriers. The authors can also make use of: 10.1016/j.ijbiomac.2017.10.170
Author Response
Manuscript ID: pharmaceutics-1772803
Title: New Advances in Polymeric Micelles for Therapeutic Applications
Dear Editor,
Doctor Yoona Xu
The authors appreciate the careful review and greatly acknowledge the comments that the Reviewer has provided on our manuscript. We have carefully revised the manuscript and have made the recommended changes and answered in detail to the questions raised. Additional information was also added when appropriate. All changes made to the text are highlighted in pink colour and marked up using the “Track Changes” function.
# Reviewer 4
This review article summarizing the new advances and therapeutic applications of polymeric miclles is a very good work that deserves publication after some improvement:
1- In page 8: the structure of the Pluronics should be corrected as they are are amphiphilic tri-block copolymers [Polyethylenoxide(PEO)a- Polypropylenoxyde(PEO)b- Polyethylenoxide(PEO)a]; with the letter "y" not "i".
Response: The authors thank for the noticing of this error. The lapse was corrected, accordingly. Please check line 231 and following.
2- Also the authors should mention that the Pluronics carry another name which is poloxamers that should be mentioned in the text.
Response: The reference to this fact was added to the manuscript. Please check lines 227 and following.
3- In section 6 dealing with the use of pluronics in cancer therapy, the authors should refer to: 10.1016/j.molliq.2015.12.007, as it mentions the role of pluronics in increasing the cytotoxicity against HEP-G2 cells.
Response: The authors thank the suggestion. The reference was added, accordingly.
4- Section 7 needs more examples for the use of polymeric micelles in topical and transdermal delivery and the suggested penetration pathways of these polymeric carriers. The authors can also make use of: 10.1016/j.ijbiomac.2017.10.170
Response: The reference was added and more examples of polymeric micelles in topical and transdermal delivery are provided. Please check lines 321 and following.
Round 2
Reviewer 1 Report
The authors of the manuscript entitled New Advances in Biomedical Application of Polymeric Micelles addressed the most of my comments appropriate. After revision, the quality of the manuscript raised significantly. Therefore, I would like to suggest the editor to accept this manuscript.
Reviewer 2 Report
Authors has revised the manuscript reasonably. It is acceptable in the present form.